# Predictors of self-reported symptoms and testing for COVID-19 in Canada using a nationally representative survey

Daphne C. Wu[ORCID]1, Prabhat Jha[ORCID]1*, Teresa Lam2, Patrick Brown[ORCID]1,3, Hellen Gelband1, Nico Nagelkerke1, H. Chaim Birnboim1,4, Angus Reid2, on behalf of the Action to Beat Coronavirus in Canada/Action pour Battre le Coronavirus (Ab-C) Study Group¶

1 Centre for Global Health Research, St. Michael's Hospital, Unity Health Toronto, and Dalla Lana School of Public Health, University of Toronto, Toronto, Ontario, Canada, 2 Angus Reid Institute, Vancouver, British Colombia, Canada, 3 Department of Statistical Sciences, University of Toronto, Toronto, Ontario, Canada, 4 deltaDNA Biosciences, Inc., Toronto, Ontario, Canada

¶ Membership of the Action to Beat Coronavirus in Canada/Action pour Battre le Coronavirus (Ab-C) Study Group is listed in the Acknowledgments.
* Prabhat.jha@utoronto.ca

**Data Availability Statement:** The authors are unable to share a de-identified dataset of the survey used in our study due to ethical and legal restrictions, as the data contain potentially

## Abstract

Random population-based surveys to estimate prevalence of SARS-CoV2 infection causing coronavirus disease (COVID-19) are useful to understand distributions and predictors of the infection. In April 2020, the first-ever nationally representative survey in Canada polled 4,240 adults age 18 years and older about self-reported COVID experience in March, early in the epidemic. We examined the levels and predictors of COVID symptoms, defined as fever plus difficulty breathing/shortness of breath, dry cough so severe that it disrupts sleep, and/or loss of sense of smell; and testing for SARS-CoV-2 by respondents and/or household members. About 8% of Canadians reported that they and/or one or more household members experienced COVID symptoms. Symptoms were more common in younger than in older adults, and among visible minorities. Overall, only 3% of respondents and/or household members reported testing for SARS-CoV-2. Being tested was associated with having COVID symptoms, Indigenous identity, and living in Quebec. Periodic nationally representative surveys of symptoms, as well as SARS-CoV-2 antibodies, are required in many countries to understand the pandemic and prepare for the future.

## Introduction

The pandemic of SARS-CoV-2 infection causing coronavirus disease-2019 (COVID-19, hereafter "COVID") has affected almost every country globally [1]. Understanding the socio-demographic characteristics of COVID patients can inform efforts to reduce transmission [2]. In Canada and other high-income country settings where reliable data can be gathered, a combination of population-based surveys (including surveys and testing), hospitalizations, and mortality data can produce an accurate profile of the impact of the pandemic.

identifying participant information. The dataset can be made available upon request to Angus Reid Institute via email at angus@angusreid.org.

**Funding:** PJ received funding from the Bill and Melinda Gates Foundation (https://www.gatesfoundation.org; grant number: OPP1159622) and the Canadian Institutes of Health Research (https://cihr-irsc.gc.ca/e/193.html; grant number: FDN 154277), and AR is funded by the Angus Reid Institute (http://angusreid.org/). The Angus Reid Institute collected the data used in this study and reviewed the manuscript. The funders (Bill and Melinda Gates Foundation and Canadian Institutes of Health Research) provided support in the form of salaries for authors (DCW, HG), but did not have any additional role in the study design, data collection and analysis, decision to public, or preparation of the manuscript. The specific roles of these authors are articulated in the 'author contributions' section.

**Competing interests:** The authors of this manuscript have read the journal's policy and have the following competing interests: HCB is a paid employee of deltaDNA Biosciences, Inc., but the company did not fund the study. There are no patents, products in development or marketed products associated with this research to declare. This does not alter our adherence to PLOS ONE policies on sharing data and materials.

Here, we report on the results of the first nationally-representative poll in Canada of self-reported COVID symptoms conducted by the Angus Reid Forum in early April 2020 covering symptoms reported mostly in March 2020, prior to the peak month of test-reported cases in April. Specifically, we seek to understand the distribution and predictors of Canadians reporting possible COVID symptoms and testing for SARS-CoV-2 using the current standard (polymerase chain reaction or PCR-based) test. We discuss these findings in the context of the age distribution of COVID hospitalizations and deaths, and a planned survey of antibodies to SARS-CoV-2 in a random sample of Canadians.

## Materials and methods

### Study design

The Angus Reid Institute (ARI) conducted an online survey from April 1–5, 2020, among a nationally representative randomized sample of 4,240 Canadian adults who are members of Angus Reid Forum, drawing upon 70,000 adults in a distributed online panel used for policy research by public sector, not-for-profit, media and commercial organizations (with sampling units approximately corresponding to each federal riding) [3]. A probability sample of this size carries a margin of error of +/- 2 percentage points, 19 times out of 20. The sample frame was established to match the Canadian census data from Statistics Canada [4]. The survey was commissioned and paid for by ARI. Respondents were required to be 18 years or older and to speak English or French.

Overall, the survey respondents were broadly representative of Canadian society in terms of gender, age, regional distribution, and numbers of household members (Table 1). Survey respondents were less representative of Canada in terms of ethnicities other than Indigenous. The survey had fewer single-member households than in the Canadian census, and had slightly higher education levels than did the 2019 Canadian population.

### Survey questionnaire and data collection

The survey questionnaire included questions related to COVID and socio-demographic characteristics among both the respondents and members of their households. Questions related to COVID symptoms included whether the respondent experienced any of a list of eight symptoms: i) difficulty breathing or shortness of breath, ii) a fever, iii) a mild dry cough, iv) a severe dry cough so severe that it disrupts sleep, v) sore throat, vi) frequent sneezing, vii) loss of sense of smell, viii) fever with hallucinations over the past month. Questions related to testing for COVID included whether the respondent have been tested for COVID, scheduled to be tested, trying to get tested but have not been able to, have done a self-assessment through government website/app, or have not been tested nor planning to be tested or self-assessed. Respondents living in households with more than one person were also asked if anyone else in the household had experienced any of the COVID symptoms and have been tested for COVID. The questionnaire used for this study was developed based on expert opinion, and the choice of COVID symptoms were based on expert opinion and physicians from Unity Health Toronto. The full survey questionnaire is provided in the (S1 Appendix). The survey instrument was pre-tested in 60 individuals prior to the main survey.

The data were collected by Angus Reid Institute and were analysed at Angus Reid Institute and Centre for Global Health Research, Unity Health Toronto. All personal identifiers were removed from the collected data and each participant was then assigned a randomly-generated code identifier before analysis.

**Table 1. Socio-demographic characteristics of respondents (n = 4,240) as compared to the Canadian population in 2019.**

| Characteristics | Completed survey, sample size | | Canadian population, 2019[*] (%) [4, 5] |
|---|---|---|---|
| | n | % | |
| **Gender** | | | |
| Male | 2017 | 47.9 | 49.4 |
| Female | 2205 | 52.0 | 50.6 |
| Other[†] | 18 | 0.4 | |
| **Age group** | | | |
| 18–45 years | 2007 | 47.3 | 46.4 |
| 46–65 years | 1511 | 35.6 | 33.4 |
| 66+ years | 722 | 17.0 | 20.2 |
| **Education** | | | |
| High school and under | 1064 | 25.1 | 35.2 |
| Some college/university and higher | 3176 | 74.9 | 64.8 |
| **Visible minority[‡]** | | | |
| No | 3686 | 86.9 | 71.4 |
| Yes | 554 | 13.1 | 28.7 |
| **Number of household members** | | | |
| Lived alone | 672 | 15.9 | 28.2 |
| Two people | 1615 | 38.1 | 34.4 |
| Three people | 804 | 19.0 | 15.2 |
| Four or more people | 1149 | 27.1 | 22.2 |
| **Ethnicity** | | | |
| Indigenous[§] | 226 | 5.3 | 5.0 |
| English and other European | 3567 | 84.1 | 73.1 |
| Others[|] | 365 | 8.6 | 22.0 |
| Rather Not Say | 82 | 1.9 | |
| **Province/region** | | | |
| Ontario | 1600 | 37.7 | 38.9 |
| British Columbia | 554 | 13.1 | 13.8 |
| Quebec | 1020 | 24.1 | 22.7 |
| Alberta | 475 | 11.2 | 11.2 |
| Manitoba | 150 | 3.5 | 3.5 |
| Saskatchewan | 133 | 3.1 | 3.0 |
| Atlantic provinces | 308 | 7.3 | 6.6 |

[*]All comparisons are from the 2016 Census, except Province/region which is from 2019 Statistics Canada population data [4].

[†]"Other" gender includes genders that neither male nor females.

[‡]Visible minority is defined as persons, other than Indigenous peoples, who are non-white in race or non-white in colour) [6].

[§]Indigenous populations including First Nations, Inuit, or Métis.

[|]"Others" ethnicity includes Caribbean, Central or South American, African, Middle Eastern, Central Asian, Chinese, Filipino, South Asian, other Asian, and Oceania.

## Analysis

To understand the socio-demographic predictors of COVID symptoms, we conducted a logistic regression analysis where the outcome was self-reported symptoms suggestive of COVID infection which we defined in this study as the respondent reporting himself/herself and/or at least one member of the household having had a combination of fever (with or without hallucinations) <u>and</u> any of i) difficulty breathing/shortness of breath <u>or</u> ii) dry cough so severe that it disrupts sleep <u>or</u> iii) a loss of a sense of smell in the past month; and the explanatory variables were gender (male, female, or other), education level (high school and under, or some college/university and higher), province, age, ethnicity (Indigenous, English and other European, or others), visible minority (defined as persons, other than Aboriginal peoples, who are non-white in race or colour) [6], and number of household members. We defined Indigenous ethnicity as whether the person reported identifying with the Aboriginal peoples of Canada which includes First Nations, Métis or Inuit and/or those who reported Registered or Treaty Indian status [7]. We also used logistic regression analysis to identify predictors of testing, which included the above explanatory variables and also COVID symptoms in the respondent or a household member. Respondents were considered "tested" if he/she had been tested or were scheduled to be tested. Respondents who did not report on any of the explanatory variables were excluded from these analyses. Results of the regression analyses were presented after adjusting for possible confounding effect of other explanatory variables as adjusted odds ratios (OR) with 95% confidence intervals (CI). Discrepancies in or between totals are due to rounding. Since the sample frame matched the Canadian census data from Statistics Canada, no additional survey weights were applied. We used RStudio Version 1.1.453 for analyses.

## Ethics approval

The Angus Reid Forum obtained consent from all participants. The data without any personal identifiers are made available openly to bona-fide researchers. IRB approval for this study was obtained from Unity Health Toronto Research Ethics Board (REB# 20–107).

## Results

Of the 4,240 respondents, 334 (or 7.9%) reported either themselves or at least one of the household members having COVID symptoms. Of these, 210 (5.0%) reported COVID symptoms only in themselves. The adjusted OR of the respondent having symptoms when at least one of the household members reported symptoms was 1.45 (95% CI 1.41–1.49); the OR was similar for at least one household member having symptoms if the respondent reported symptoms. In terms of testing for SARS-CoV-2, 126 (or 3.0%) reported some household testing or being scheduled for testing, and only 68 (or 1.6%) reported that they have been or are scheduled for testing. Details of the variation in COVID symptoms and SARS-CoV-2 testing in this sample across provinces have already been published [3].

Table 2 shows the adjusted OR of respondents or a member of the household, and respondents only having COVID symptoms after adjustment for other variables examined. We excluded 99 (or 2.3%) respondents who did not report on at least one of the variables. The proportion of respondents reporting COVID symptoms within the household decreased with age: 11.2% of those aged 18–45 years, 5.6% among those aged 46–65 years, and 3.1% among those aged 66 years and older. The lower prevalence at higher ages was similar among those reporting COVID symptoms only themselves. After controlling for gender, province, age, ethnicity, visible minority, and number of household members, older adults were significantly less likely to report having COVID symptoms themselves or within the household (age 46–65, OR = 0.55, 0.42–0.71; age 66+, OR = 0.30, 0.18–0.47). Those who reported themselves as a

**Table 2. Logistic regression models for respondents or a member of household, and respondents only having COVID symptoms.**

| Characteristics | Respondents or a member of household having symptoms | | | Respondents only having symptoms | | |
|---|---|---|---|---|---|---|
| | COVID symptoms present/absent | % | Adjusted odds ratios (95% CI) | COVID symptoms present/absent | % | Adjusted odds ratios (95% CI) |
| | N = 324/3817 | | | N = 202/3939 | | |
| **Gender** | | | | | | |
| Male | 149/1840 | 7.5 | Ref | 85/1904 | 4.3 | Ref |
| Female | 175/1977 | 8.1 | 1.11 (0.89–1.40) | 117/2036 | 5.4 | 1.30 (0.97–1.74) |
| **Age group** | | | | | | |
| 18–45 years | 218/1724 | 11.2 | Ref | 126/1816 | 6.5 | Ref |
| 46–65 years | 84/1403 | 5.6 | **0.55 (0.42–0.71)** | 61/1426 | 4.1 | **0.61 (0.44–0.84)** |
| 66+ years | 22/690 | 3.1 | **0.30 (0.18–0.47)** | 15/697 | 2.1 | **0.29 (0.16–0.50)** |
| **Education** | | | | | | |
| Some college/university and higher | 237/2874 | 8.4 | Ref | 150/2961 | 4.8 | Ref |
| High school and under | 87/943 | 7.6 | 1.25 (0.96–1.62) | 52/978 | 5.1 | 1.16 (0.83–1.61) |
| **Visible minority** | | | | | | |
| No | 256/3359 | 7.1 | Ref | 155/3460 | 4.3 | Ref |
| Yes | 68/458 | 12.9 | **1.55 (1.08–2.20)** | 47/479 | 8.9 | **2.08 (1.34–3.15)** |
| **Number of household members** | | | | | | |
| Two people | 108/1479 | 6.8 | Ref | 79/1507 | 5.0 | Ref |
| Lived alone | 29/626 | 4.4 | 0.69 (0.45–1.02) | 29/626 | 4.5 | 1.01 (0.64–1.53) |
| Three people | 77/704 | 9.9 | 1.17 (0.85–1.61) | 45/736 | 5.8 | 0.86 (0.57–1.28) |
| Four or more people | 110/1008 | 9.8 | 1.19 (0.89–1.59) | 48/1070 | 4.3 | 0.72 (0.49–1.05) |
| **Ethnicity** | | | | | | |
| English and other European | 252/3305 | 7.1 | Ref | 158/3399 | 4.4 | Ref |
| Indigenous* | 27/195 | 12.2 | 1.31 (0.83–2.01) | 19/203 | 8.5 | 1.34 (0.76–2.24) |
| Others† | 45/317 | 12.4 | 1.20 (0.79–1.80) | 25/337 | 7.0 | 0.87 (0.50–1.47) |
| **Province/region** | | | | | | |
| Ontario | 129/1448 | 8.2 | Ref | 83/1493 | 5.3 | Ref |
| British Columbia | 44/495 | 8.2 | 1.08 (0.77–1.51) | 23/516 | 4.2 | 0.80 (0.51–1.24) |
| Quebec | 65/922 | 6.6 | 0.93 (0.66–1.29) | 42/945 | 4.2 | 0.85 (0.56–1.27) |
| Prairie provinces | 63/674 | 8.5 | 1.09 (0.79–1.49) | 39/697 | 5.2 | 0.98 (0.66–1.45) |
| Atlantic provinces | 23/278 | 7.6 | 1.11 (0.68–1.76) | 15/287 | 5.0 | 0.99 (0.53–1.74) |

*Indigenous populations including First Nations, Inuit, or Métis.

†"Others" ethnicity includes Caribbean, Central or South American, African, Middle Eastern, Central Asian, Chinese, Filipino, South Asian, other Asian, and Oceania.

Bolded values indicate significance at 95% confidence interval. Adjustment was for the other variables in the table.

The number reporting COVID symptoms was 341 in unweighted analyses and 334 after weighing the sample to the Canadian census data, showing the sampling frame as robust.

visible minority were significantly more likely to have COVID symptoms (12.9%) compared to those who did not (7.1%; adjusted OR = 1.55; 1.08–2.20). Similar results were found when we examined COVID symptoms only among respondents. The associations changed substantially between unadjusted and adjusted analyses, most notably for ethnicity, and somewhat for being a visible minority, suggesting that some residual confounding factors were present.

Table 3 shows the adjusted OR of respondents or a member of the household being tested or scheduled for SARS-CoV-2 testing. The strongest predictor of testing was having COVID symptoms among members of the household, of whom about 16.5% were tested, compared to 2.1% among those without COVID symptoms among household members (OR = 6.63, 4.46–

**Table 3. Logistic regression models for respondent or a member of household having been tested for SARS-CoV-2.**

| Characteristics | Respondents or a member of household having been tested | | | Respondents only having been tested | | |
|---|---|---|---|---|---|---|
| | SARS-CoV-2 tested/not tested | | Adjusted odds ratios (95% CI) | SARS-CoV-2 tested/not tested | | Adjusted odds ratios (95% CI) |
| | N = 126/4015 | % | | N = 66/4075 | % | |
| **Respondent and/or household member had COVID symptoms** | | | | | | |
| No | 80/3737 | 2.1 | Ref | 42/3775 | 1.1 | Ref |
| Yes | 46/278 | 16.5 | **6.63 (4.46–9.79)** | 24/300 | 7.5 | **6.63 (3.82–11.32)** |
| **Gender** | | | | | | |
| Male | 68/1921 | 3.5 | Ref | 34/1955 | 1.7 | Ref |
| Female | 58/2094 | 2.7 | 0.72 (0.51–1.03) | 32/2120 | 1.5 | 0.85 (0.52–1.38) |
| **Age group** | | | | | | |
| 18–45 years | 84/1858 | 4.5 | Ref | 41/1900 | 2.1 | Ref |
| 46–65 years | 32/1455 | 2.2 | 0.69 (0.45–1.05) | 19/1468 | 1.3 | 0.75 (0.42–1.31) |
| 66+ years | 10/702 | 0.7 | 0.54 (0.25–1.05) | 5/707 | 0.7 | 0.44 (0.14–1.08) |
| **Education** | | | | | | |
| Some college/university and higher | 96/3015 | 3.1 | Ref | 52/3059 | 1.7 | Ref |
| High school and under | 29/1000 | 2.8 | 0.88 (0.56–1.35) | 14/1016 | 1.3 | 0.74 (0.37–1.35) |
| **Visible minority** | | | | | | |
| No | 96/3519 | 2.7 | Ref | 52/3563 | 1.4 | Ref |
| Yes | 30/496 | 5.7 | 0.83 (0.36–1.05) | 14/512 | 2.7 | 0.54 (0.27–1.15) |
| **Number of household members** | | | | | | |
| Two people | 38/1548 | 2.4 | Ref | 19/1567 | 1.2 | Ref |
| Lived alone | 13/642 | 2.0 | 0.83 (0.43–1.52) | 13/642 | 2.0 | 1.77 (0.86–3.57) |
| Three people | 28/754 | 3.6 | 1.18 (0.69–1.97) | 18/764 | 2.3 | 1.28 (0.62–2.56) |
| Four or more people | 47/1071 | 4.2 | 1.32 (0.84–2.09) | 16/1102 | 1.4 | 0.93 (0.47–1.84) |
| **Ethnicity** | | | | | | |
| English and other European | 95/3462 | 2.7 | Ref | 54/3503 | 1.5 | Ref |
| Indigenous* | 16/206 | 7.2 | **2.07 (1.13–3.64)** | 9/213 | 3.9 | 2.04 (0.89–4.28) |
| Others† | 15/347 | 4.1 | 0.99 (0.49–1.89) | 4/359 | 1.0 | 0.47 (0.13–1.32) |
| **Province/region** | | | | | | |
| Ontario | 40/1537 | 2.5 | Ref | 22/1554 | 1.4 | Ref |
| British Columbia | 15/524 | 2.8 | 1.08 (0.59–1.93) | 7/532 | 1.3 | 0.81 (0.34–1.81) |
| Quebec | 47/940 | 4.8 | **2.41 (1.49–3.96)** | 26/961 | 2.6 | **1.98 (1.05–3.81)** |
| Prairie provinces | 19/717 | 2.6 | 1.13 (0.65–1.95) | 7/730 | 1.0 | 0.74 (0.32–1.63) |
| Atlantic provinces | 5/297 | 1.7 | 0.92 (0.34–2.14) | 4/299 | 1.2 | 1.03 (0.29–2.87) |

*Indigenous populations including First Nations, Inuit, or Métis.

†"Others" ethnicity includes Caribbean, Central or South American, African, Middle Eastern, Central Asian, Chinese, Filipino, South Asian, other Asian, and Oceania. Bolded values indicate significance at 95% confidence interval. Adjustment was for the other variables in the table.

9.79). Testing rates fell with age, but not significantly so. Testing rates were 2.7% in English and other European ethnicity, and higher in those of indigenous ethnicity (7.2%; OR = 2.07; 1.13–3.64). However, the ORs fell notably (from 2.94 to 2.07) after adjustment for co-variates, suggesting residual confounding. Testing rates in Quebec were roughly double those in Ontario, the reference province (OR = 2.41; 1.49–3.96).

## Discussion

A nationally representative survey of Canadians finds that about 8% of adults report that they or someone in their household reported symptoms suggestive of COVID in March 2020. Being a visible minority was associated with higher self-reported COVID symptoms. Self-reported symptoms were notably less common at older ages than in younger adults. Only 3% of Canadian adults reported that they or someone in the household had been tested for SARS-CoV-2. The main predictors of being tested were the presence of COVID symptoms, being of Indigenous identity, and living in Quebec. Testing rates were somewhat lower in older adults. There are surprisingly few national studies, and despite some limitations, this study represents the first to document self-reported symptoms in a reasonably representative sample.

COVID symptoms overlap with some other infections, notably seasonal influenza, which could have inflated the coronavirus rates in this survey. However, a study comparing the COVID symptom syndrome with other infections in the United States suggests that most are actually due to COVID [8]. We found possible COVID symptoms to be less prevalent and the levels of testing marginally less prevalent in older adults than other age groups, despite the certainty that the vast majority of COVID hospitalizations and deaths occur at older ages. However, a weakness of our sample is the lack of representation from nursing and long-term care residents, in whom more than three-quarters of all COVID deaths occur [1]. Approximately 0.3 million (out a total population of 38 million) Canadians are estimated to live in nursing homes/long-term care institutions [9]. This would represent less than 1% of the expected population in the survey. Such populations were under-represented in the survey. There may be additional reasons for the age-specific findings, however. Anecdotal reports suggest that older adults do not experience the symptoms used to define COVID infection in the poll, but may report vaguer symptoms such as dizziness and confusion [10]. Further surveys focused on syndromes that might occur in older adults are warranted. The testing results are broadly consistent with reports of the general levels of access to SARS-CoV-2 testing during the survey time period, including a higher level of testing in Quebec than in Ontario. However, testing results tend not to be representative of the population [3].

This syndromic survey provides some insights into the prevalence of actual SARS-CoV-2 in Canada. This time period of symptoms in March is consistent with testing data showing a peak exposure and incidence of COVID occurred in mid- to late April, presumably about 10–14 days in those adults who were tested because of their symptoms [1]. A Forum Research and Mainstreet Research reported an estimated prevalence of 5–8% in early-to-mid April in Ontario, which is consistent with our overall results for that province. However, they used a narrow range of symptoms from those we defined in our study [11].

In the UK, pilot results from the COVID-19 Infection Survey being carried out by the Office for National Statistics found that 0.25% of the community population in England above the age of 2, tested positive for the SARS-CoV-2 antigen in early-to-mid May 2020. Testing involved home self-tests of nasal swabs, and excluded those in hospitals, care homes, or other institutional settings [12]. The prevalence was lower than expected in the pilot, and larger studies are planned, along with antibody surveys. A nationally representative sero-epidemiological study is needed to establish the population distribution of cumulative SARS-CoV-2 infection, which would capture both symptomatic and asymptomatic cases. Such studies have now been conducted in England [13], Iceland [14], Brazil [15], and Spain [16]. We have begun such a study in Canada called Action to Beat Coronavirus/Action pour Battre le Coronavirus (Ab-C). The study protocol is provided in the (S2 Appendix). This study will determine the cumulative prevalence of SARS-CoV-2 infection during March-May 2020 in Canada through testing for IgG antibodies (using dual assays), paired with household questionnaire data on COVID

experience. A second round of questionnaires and antibody testing four to six months later in the same individuals will provide information on ongoing transmission, changes in the immune status of the population, and the durability of the immune response. The Angus Reid Forum is partnering with the Centre for Global Health Research at Unity Health Toronto and the University of Toronto on this study. The expected sample frame is 14,000 individuals surveyed, of which we expect about 8,000 to agree to provide a blood sample.

The present analysis informs the design of antibody surveys. For example, since the strongest predictor of COVID testing was the presence of symptoms, a similar higher prevalence of antibodies might be expected among those reporting symptoms (who may be more likely to enroll in the survey). Thus, ensuring that the sample size is sufficiently large to examine the seroprevalence in those without symptoms is key. Similarly, while the current study did not examine geographic clustering of COVID symptoms, this might well be the case with actual SARS-CoV-2 infection. There have been calls for use of self-reported syndromic data to monitor the epidemic [17]. We believe, that these strategies should include also large and as diversely representative survey as possible. Older adults bear the brunt of COVID hospitalization and deaths. However, this population reported a lower prevalence of symptoms than younger adults. Thus, the Ab-C study will oversample the population age 60 or older, particularly to understand the possible role of asymptomatic infections at older ages. As the Ab-C study is not likely to capture the prevalence of infection among nursing home residents, ancillary studies are needed to quantify hazards among this important group.

## Limitations

Self-reported symptoms are, by their nature, subject to limitations and misclassification. However, the trends over time, even with misclassification, are useful for understanding trends in the actual underlying prevalence of infection. This is because the "noise" of COVID symptoms should change little from one survey to the next, and provided there is large "signal" due to the COVID pandemic—certainly the case with this virus—reliable estimation of the trends of infection are possible. Similar methodological insights have arisen from HIV-1 testing in pregnant populations in various countries [18]. The biases in enrollment are also inherent in any polling, but the Angus Reid polling showed reasonably good consistency with the Canadian 2016 census. Moreover, the testing results did not appear to be biased greatly, and the more common testing reported in Quebec is consistent with public health reports. Another limitation of the study is that the survey questionnaire was not validated against serological results as none were available at the time of study design. However, subsequent studies comparing antibody with symptoms which we defined as suggestive of COVID in our study confirmed that our choice of symptoms was appropriate [19].

In conclusion, COVID surveillance using nationally representative surveys is essential, and ideally needs to be accompanied by antibody determination of infection. Particular attention must be paid to symptoms and testing levels in older adults.

## Supporting information

**S1 Appendix. COVID-19 questionnaire (Angus Reid Institute, wave 6).**
(DOCX)

**S2 Appendix. The Action to Beat Coronavirus/Action pour Battre le Coronavirus (Ab-C) study protocol.**
(DOCX)

## Acknowledgments

The Action to Beat Coronavirus in Canada/Action pour Battre le Coronavirus (Ab-C) Study Group, by institution: St. Michael's Hospital, Unity Health Toronto and University of Toronto (Prabhat Jha, Heyu Ni, Arthur Slutsky, Gillian Booth, Patrick Brown, Peter Juni, Hellen Gelband, Nico Nagelkerke, Abha Sharma, Peter Rodriguez, Craig Schultz, Daphne Wu, Xuyang Tang, Divya Raman Santhanam, Rajeev Kamadod, Vedika Jha); St. Joseph's Hospital, Unity Health Toronto and University of Toronto (Maria Pasic, Ron Weingust); Sinai Health, Toronto (Anne-Claude Gingras, Karen Colwill); deltaDNA Bioscience, Inc (H. Chaim Birnboin); University Health Network (Isaac Bogoch, Rupert Kaul); Children's Hospital of Eastern Ottawa (Pranesh Chakraborty); Angus Reid Institute (Angus Reid, Ed Morawski, Demetre Eliopoulos, Andy Hollander, Teresa Lam).

Lead author: Prabhat Jha, email: Prabhat.Jha@utoronto.ca.

## Author Contributions

**Conceptualization:** Prabhat Jha, Angus Reid.

**Data curation:** Daphne C. Wu.

**Formal analysis:** Daphne C. Wu, Teresa Lam.

**Funding acquisition:** Prabhat Jha.

**Investigation:** Daphne C. Wu, Prabhat Jha, Patrick Brown.

**Methodology:** Daphne C. Wu, Prabhat Jha, Angus Reid.

**Project administration:** Prabhat Jha.

**Resources:** Prabhat Jha, Angus Reid.

**Supervision:** Prabhat Jha.

**Validation:** Daphne C. Wu, Prabhat Jha, Nico Nagelkerke, H. Chaim Birnboim, Angus Reid.

**Writing – original draft:** Daphne C. Wu, Prabhat Jha.

**Writing – review & editing:** Daphne C. Wu, Prabhat Jha, Patrick Brown, Hellen Gelband, Nico Nagelkerke, H. Chaim Birnboim, Angus Reid.

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
