## [Decision Letter · Decision Letter 0]

6 Jul 2020

PONE-D-20-16713

Determinants of self-reported symptoms and testing for COVID-19 in Canada using a nationally representative survey

PLOS ONE

Dear Dr. Jha,

Thank you for submitting your manuscript to PLOS ONE. After careful consideration, we feel that it has merit but does not fully meet PLOS ONE’s publication criteria as it currently stands. Therefore, we invite you to submit a revised version of the manuscript that addresses the points raised during the review process.

Abstract to be structured as per PLOS ONE policy

Introduction should be more explicit, authors may wish to use pubmed https://pubmed.ncbi.nlm.nih.gov/?term=Determinants+of+self-reported+symptoms+and+testing+for+COVID-19 and other literature search platform.

There is a need to describe the outcome variables and the independent variables including plausible covariates, this will assist reader to understand the variables of interests for determinant of self-reported symptoms and testing for COVID-19.

A bit of explanation on how data was managed to de-identify and assign unique identifier.

The statement of ethics should be very clear and the waiver reference number should be provided.

Include the statement for authors contribution

We look forward to receiving your revised manuscript.

Kind regards,

Olanrewaju Oladimeji, Ph.D., MB; BS

Academic Editor

PLOS ONE

Journal Requirements:

3. Please clarify your Data availability statement. Please note that PLOS ONE criteria require that authors make all data underlying the findings described in their manuscript fully available without restriction at the time of publication. Authors should share any data specific to their analysis that they can legally distribute. For data that they cannot legally distribute, the authors must include in the Data Availability Statement all necessary contact information where an interested researcher would need to apply to gain access to the data; we do not allow an author to be the only point of contact for fielding requests for access to restricted data (for more information, please see https://journals.plos.org/plosone/s/data-availability). 

We note that you have indicated that data from this study are available upon request. PLOS only allows data to be available upon request if there are legal or ethical restrictions on sharing data publicly. For information on unacceptable data access restrictions, please see http://journals.plos.org/plosone/s/data-availability#loc-unacceptable-data-access-restrictions.

4. Please confirm whether a waiver of Ethics approval was obtained."  

5. In your Methods section, please provide additional information about the participant recruitment method and the demographic details of your participants. Please ensure you have provided sufficient details to replicate the analyses such as: a) the recruitment date range (month and year), b) a description of any inclusion/exclusion criteria that were applied to participant recruitment, c) a table of relevant demographic details, d) a statement as to whether your sample can be considered representative of a larger population, e) a description of how participants were recruited, and f) descriptions of where participants were recruited and where the research took place. Furthermore, please provide more information on the survey used. in the methods section, please refer specifically to the questions used to assess variables; discuss how items (for example, the choice of COVID-19 symptoms) were originated; and  if you developed the questionnaire as part of this study and it is not under a copyright more restrictive than CC-BY, please include a copy, in both the original language and English, as Supporting Information. Moreover, please include more details on how the questionnaire was pre-tested, and whether it was validated. "

6. Please note that according to our submission guidelines (http://journals.plos.org/plosone/s/submission-guidelines), outmoded terms and potentially stigmatizing labels should be changed to more current, acceptable terminology. For example: “Caucasian” should be changed to “white” or “of [Western] European descent

7. Thank you for stating the following in the Financial Disclosure section:

"PJ received funding from the Canadian Institutes of Health Research (https://cihrirsc.

gc.ca/e/193.html; grant number: FDN 154277), and AR is funded by the Angus

Reid Institute (http://angusreid.org/). The Angus Reid Institute collected the data used

in this study and reviewed the manuscript."

We note that one or more of the authors are employed by a commercial company: "deltaDNA Biosciences, Inc.,".

8. One of the noted authors is a group or consortium "Action to Beat Coronavirus in Canada/Action pour Battre le Coronavirus (Ab-C) Study Group". In addition to naming the author group, please list the individual authors and affiliations within this group in the acknowledgments section of your manuscript. Please also indicate clearly a lead author for this group along with a contact email address.

Additional Editor Comments (if provided):

Abstract to be structured as per PLOS ONE policy

Introduction should be more explicit, authors may wish to use Pubmed https://pubmed.ncbi.nlm.nih.gov/?term=Determinants+of+self-reported+symptoms+and+testing+for+COVID-19 and other literature search platform.

There is a need to describe the outcome variables and the independent variables including plausible covariates, this will assist reader to understand the variables of interests for determinant of self-reported symptoms and testing for COVID-19.

A bit of explanation on how data was managed to de-identify and assign unique identifier.

The statement of ethics should be very clear and the waiver reference number should be provided.

Include the statement for authors contribution

Reviewers' comments:

Reviewer's Responses to Questions

**Comments to the Author**

1. Is the manuscript technically sound, and do the data support the conclusions?

Reviewer #1: Yes

Reviewer #2: Yes

2. Has the statistical analysis been performed appropriately and rigorously? 

Reviewer #1: I Don't Know

Reviewer #2: Yes

3. Have the authors made all data underlying the findings in their manuscript fully available?

Reviewer #1: Yes

Reviewer #2: Yes

4. Is the manuscript presented in an intelligible fashion and written in standard English?

Reviewer #1: Yes

Reviewer #2: Yes

5. Review Comments to the Author

Reviewer #1: REVIEWER COMMENT

The authors have presented a self-reported symptom and testing sampling of COVID-19 in Canada. Though a survey polling of this nature could come with some anecdotal exaggerations and biases, the study shows that the polling represents to a greater extent the Canadian census demographics.

MINOR COMMENTS

- The authors reported lower self-reported symptoms amongst the older age group, which they justified by the fact of differential presentation with dizziness and confusion, unlike the major listed symptoms for COVID.

- In addition, the authors reported “living in Quebec” (See line 133) as a determinant of “being tested”. It should be noted that Quebec has been the hardest hit province and this presupposes that the testing rates ought to be higher. If the authors want to make the assertion that “living in Quebec” is a determinant of “being tested”, they should try and run a regression with the testing rates and the number of reported COVID positive cases across the provinces when this data was collected. Otherwise, the authors may decide not to include that as a determinant.

Overall, the presentation and the flow are generally good for the readers to understand.

Reviewer #2: Title:

1.The author may consider revising the title to predictors of self-reported symptoms and testing for COVID 19 in Canada using a nationally representative survey. In view of the depth of the statistical analysis done.

Abstract:

1.The authors should consider including a sentence or two on COVID 19 as the opening narration of the abstract. This will bring to context the essence of the study.

Introduction:

1. The introduction will benefit from additional information on the subject matter (COVID 19) and not just the expression of what the study was all about.

Materials and Methods:

1.The authors should consider defining what constitutes the outcome variable (COVID 19) symptoms (present or absent) properly in the methodology as it was done in the abstract (COVID symptoms, defined as fever plus difficulty breathing/shortness of breath, dry cough so severe that it disrupts sleep, and/or loss of sense of smell). The abstract is expected to represent an excerpt of the content of the manuscript.

2.This manuscript will benefit from the details of how the survey weights was applied to the prevalence. Though a reference was provided but the cited document does not provide the needed clarity.

3.More information has to be provided by the authors on why IRB was not required for the study of this magnitude. It is also unclear what the statement "as per Unity Health Toronto Research Ethics Board" means. Was the ethics approval waived by unity Toronto Research Board? if so, reasons for such waiver have to be provided.

Results:

1. The information on the representativeness of the respondents by comparing the demographic characteristics with the Canadian population of 2019 will be more appropriate in the method as a standardization measure built into the study rather than having it in the results section.

2.Table 1; it is unclear what gender constitutes others in the absence of an appropriate footnote.

3. The authors should consider expunging the information on part of lines 133 to 115" Results using a narrower definition of COVID symptoms, namely having fever, difficulty breathing/shortness of breath, and severe dry cough were similar (data not shown) " as it is not adding any credence to the study.

4.Table 2; the authors may consider revising the expression used for categorizing the outcome variable (COVID symptoms) from positive/negative to present or absent. This revision may bring clarity to the fact that these symptoms were not tested for but rather self reported.

Discussion:

1.The discussion will be benefit from some revision by bringing in other similar available studies though it is a common knowledge that there is a paucity of such studies. Efforts should be made to get some of those studies available so as to enrich the discussion

6. PLOS authors have the option to publish the peer review history of their article (what does this mean?). If published, this will include your full peer review and any attached files.

Reviewer #1: **Yes: **Stanley Meribe

Reviewer #2: **Yes: **Tolulope Olumide Afolaranmi

---

## [Author Response · Author response to Decision Letter 0]

16 Jul 2020

Revision to submission [PONE-D-20-16713]: Determinants of self-reported symptoms and testing for COVID-19 in Canada using a nationally representative survey

Comment 1- Please ensure that your manuscript meets PLOS ONE's style requirements, including those for file naming. The PLOS ONE style templates can be found at https://journals.plos.org/plosone/s/file?id=wjVg/PLOSOne_formatting_sample_main_body.pdf and https://journals.plos.org/plosone/s/file?id=ba62/PLOSOne_formatting_sample_title_authors_affiliations.pdf

Response 1- We have edited the manuscript to meet PLOS ONE’s style requirements.

Comment 2- We note that you have included the phrase “data not shown” in your manuscript. Unfortunately, this does not meet our data sharing requirements. PLOS does not permit references to inaccessible data. We require that authors provide all relevant data within the paper, Supporting Information files, or in an acceptable, public repository. Please add a citation to support this phrase or upload the data that corresponds with these findings to a stable repository (such as Figshare or Dryad) and provide and URLs, DOIs, or accession numbers that may be used to access these data. Or, if the data are not a core part of the research being presented in your study, we ask that you remove the phrase that refers to these data.

Response 2- We have removed those statements with the phrase “data not shown”, as these results were not a core part of the research being presented in our study.

Comment 3- Please clarify your Data availability statement. Please note that PLOS ONE criteria require that authors make all data underlying the findings described in their manuscript fully available without restriction at the time of publication. Authors should share any data specific to their analysis that they can legally distribute. For data that they cannot legally distribute, the authors must include in the Data Availability Statement all necessary contact information where an interested researcher would need to apply to gain access to the data; we do not allow an author to be the only point of contact for fielding requests for access to restricted data (for more information, please see https://journals.plos.org/plosone/s/data-availability). 

We note that you have indicated that data from this study are available upon request. PLOS only allows data to be available upon request if there are legal or ethical restrictions on sharing data publicly. For information on unacceptable data access restrictions, please see http://journals.plos.org/plosone/s/data-availability#loc-unacceptable-data-access-restrictions.

Response 3- We have changed the Data availability statement (line 121-126).

Response- We have revised the cover letter to reflect the legal and ownership restrictions on sharing de-identified dataset.

Comment 4- Please confirm whether a waiver of Ethics approval was obtained." 

Response 4- IRB approval for this study was obtained from Unity Health Toronto Research Ethics Board (REB# 20-107).

Comment 5- In your Methods section, please provide additional information about the participant recruitment method and the demographic details of your participants. Please ensure you have provided sufficient details to replicate the analyses such as: a) the recruitment date range (month and year), 

Response 5- We have provided the recruitment date range- April 1-5, 2020. Demographic details of the participants are given in the Results section in Table 1.

b) a description of any inclusion/exclusion criteria that were applied to participant recruitment, 

Response- We have now provided the inclusion/exclusion criteria for participant recruitment (lines 62-63)

c) a table of relevant demographic details, 

Response- We have included a table of relevant demographic details as Table 1.

d) a statement as to whether your sample can be considered representative of a larger population, 

Response- We have included the statement: “The Angus Reid Institute (ARI) conducted an online survey from April 1-5, 2020, among a nationally representative randomized sample of 4,240 Canadian adults who are members of Angus Reid Forum, a national online sample of 50,000 Canadians used for political and other social polling”.

e) a description of how participants were recruited, 

Response- The sampling frame is an existing Angus Reid Forum Panel (http://www.angusreidforum.com/), which is a national online sample of 50,000 adult Canadians used for political and other social polling (lines 60-61).

f) descriptions of where participants were recruited and where the research took place

Response- We have added this in lines 58-61, 87-90.

Furthermore, please provide more information on the survey used. In the methods section, please refer specifically to the questions used to assess variables; 

Response- We have now added this in lines 95-106, 107-110.

Discuss how items (for example, the choice of COVID-19 symptoms) were originated; 

Response- We have now added this in lines 83-84.

and if you developed the questionnaire as part of this study and it is not under a copyright more restrictive than CC-BY, please include a copy, in both the original language and English, as Supporting Information. 

Response- We have now added the questionnaire as Supporting Information (S1 Appendix).

Moreover, please include more details on how the questionnaire was pre-tested, and whether it was validated. "

Response-The questionnaire was pre-tested as per standard polling procedures in about 60 people. Validation of symptoms was not possible at the time against antibody status but is planned in the main Ab-C study. We have added a line of pre-testing (lines 85-86).

Comment 6- Please note that according to our submission guidelines (http://journals.plos.org/plosone/s/submission-guidelines), outmoded terms and potentially stigmatizing labels should be changed to more current, acceptable terminology. For example: “Caucasian” should be changed to “white” or “of [Western] European descent.

Response 6- We have changed our terminologies from “Caucasian” to “white” and from “indigenous people” to “those of self-identified indigenous ethnicity”. 

Comment 7- Thank you for stating the following in the Financial Disclosure section: "PJ received funding from the Canadian Institutes of Health Research (https://cihrirsc.gc.ca/e/193.html; grant number: FDN 154277), and AR is funded by the Angus Reid Institute (http://angusreid.org/). The Angus Reid Institute collected the data used in this study and reviewed the manuscript." We note that one or more of the authors are employed by a commercial company: "deltaDNA Biosciences, Inc.,".

Response 7a-“We have updated the funders. The commercial affiliation is only that for CB and no funding was involved. We have added the statement: “The funders provided support in the form of salaries for authors [DCW, HG], but did not have any additional role in the study design, data collection and analysis, decision to publish, or preparation of the manuscript. The specific roles of these authors are articulated in the ‘author contributions’ section.”

Comment 7b- Please also provide an updated Competing Interests Statement declaring this commercial affiliation along with any other relevant declarations relating to employment, consultancy, patents, products in development, or marketed products, etc. 

Response 7b- We declare no competing interests.

Comment 8- One of the noted authors is a group or consortium "Action to Beat Coronavirus in Canada/Action pour Battre le Coronavirus (Ab-C) Study Group". In addition to naming the author group, please list the individual authors and affiliations within this group in the acknowledgments section of your manuscript. Please also indicate clearly a lead author for this group along with a contact email address.

Response 8- We have added the list of individual authors and affiliations within the group, "Action to Beat Coronavirus in Canada/Action pour Battre le Coronavirus (Ab-C) Study Group" in the Acknowledgement section, along with the lead author and contact email address. 

Additional Editor Comments

Comment 1- Abstract to be structured as per PLOS ONE policy.

Response 1- We have structured the abstract as per PLOS ONE policy.

Comment 2-Introduction should be more explicit, authors may wish to use pubmed https://pubmed.ncbi.nlm.nih.gov/?term=Determinants+of+self-reported+symptoms+and+testing+for+COVID-19 and other literature search platform.

Response 2- We have revised the Introduction to be more explicit. 

Comment 3- There is a need to describe the outcome variables and the independent variables including plausible covariates, this will assist reader to understand the variables of interests for determinant of self-reported symptoms and testing for COVID-19.

Response 3- We have now provided a description of the outcome and independent variables (lines 94-110).

Comment 4- A bit of explanation on how data was managed to de-identify and assign unique identifier.

Response 4- We have now added details on how data was managed to de-identify and assign unique identifier (lines 88-90).

Comment 5- The statement of ethics should be very clear and the waiver reference number should be provided.

Response 5- We have now added the research ethics statement and the ethics approval number (line 118-119).

Comment 6- Include the statement for authors contribution.

Response 6- We have now added the statement for authors’ contribution in the title page.

Reviewers’ comments

Reviewer #1: REVIEWER COMMENT

The authors have presented a self-reported symptom and testing sampling of COVID-19 in Canada. Though a survey polling of this nature could come with some anecdotal exaggerations and biases, the study shows that the polling represents to a greater extent the Canadian census demographics.

MINOR COMMENTS

- The authors reported lower self-reported symptoms amongst the older age group, which they justified by the fact of differential presentation with dizziness and confusion, unlike the major listed symptoms for COVID.

- In addition, the authors reported “living in Quebec” (See line 133) as a determinant of “being tested”. It should be noted that Quebec has been the hardest hit province and this presupposes that the testing rates ought to be higher. If the authors want to make the assertion that “living in Quebec” is a determinant of “being tested”, they should try and run a regression with the testing rates and the number of reported COVID positive cases across the provinces when this data was collected. Otherwise, the authors may decide not to include that as a determinant.

Response- We mention that testing rates are higher in Quebec in the discussion but do not believe it is appropriate to run a regression of testing data (which unlike this survey) are not representative versus symptom data.

Overall, the presentation and the flow are generally good for the readers to understand.

Reviewer #2: Title:

1.The author may consider revising the title to predictors of self-reported symptoms and testing for COVID 19 in Canada using a nationally representative survey. In view of the depth of the statistical analysis done.

Response 1- We have revised the title to predictors of self-reported symptoms and testing for COVID-19 in Canada using a nationally representative survey.

Abstract:

1.The authors should consider including a sentence or two on COVID 19 as the opening narration of the abstract. This will bring to context the essence of the study.

Response 1- We have now included a sentence on estimating the prevalence of COVID at the beginning of the abstract (lines 24-26).

Introduction:

1. The introduction will benefit from additional information on the subject matter (COVID 19) and not just the expression of what the study was all about.

Response 1- We have now added more information on COVID in the Introduction (lines 41-42). 

Materials and Methods:

1.The authors should consider defining what constitutes the outcome variable (COVID 19) symptoms (present or absent) properly in the methodology as it was done in the abstract (COVID symptoms, defined as fever plus difficulty breathing/shortness of breath, dry cough so severe that it disrupts sleep, and/or loss of sense of smell). The abstract is expected to represent an excerpt of the content of the manuscript.

Response 1- We have now added the definition of COVID symptoms in the Methods section (lines 75-78).

2.This manuscript will benefit from the details of how the survey weights was applied to the prevalence. Though a reference was provided but the cited document does not provide the needed clarity.

Response 2- We have now added a statement of how weightings were applied to prevalence (lines 112-113).

3.More information has to be provided by the authors on why IRB was not required for the study of this magnitude. It is also unclear what the statement "as per Unity Health Toronto Research Ethics Board" means. Was the ethics approval waived by unity Toronto Research Board? if so, reasons for such waiver have to be provided.

Response 3- We have obtained IRB approval from Unity Health Toronto (lines 118-119)

Results:

1. The information on the representativeness of the respondents by comparing the demographic characteristics with the Canadian population of 2019 will be more appropriate in the method as a standardization measure built into the study rather than having it in the results section.

Response 1- We have now moved this to the Methods section.

2.Table 1; it is unclear what gender constitutes others in the absence of an appropriate footnote.

Response 2- We have now included a footnote to clarify what “other” gender constitutes.

3. The authors should consider expunging the information on part of lines 133 to 115" Results using a narrower definition of COVID symptoms, namely having fever, difficulty breathing/shortness of breath, and severe dry cough were similar (data not shown) " as it is not adding any credence to the study.

Response 3- We have now removed the statement.

4.Table 2; the authors may consider revising the expression used for categorizing the outcome variable (COVID symptoms) from positive/negative to present or absent. This revision may bring clarity to the fact that these symptoms were not tested for but rather self reported.

Response 4- We have now changed the expression from “positive/negative” to “present/absent”.

Discussion:

1.The discussion will be benefit from some revision by bringing in other similar available studies though it is a common knowledge that there is a paucity of such studies. Efforts should be made to get some of those studies available so as to enrich the discussion.

Response 1- We have made the point about such studies being required in the discussion and cited a recent NEJM paper.

---

## [Decision Letter · Decision Letter 1]

21 Sep 2020

PONE-D-20-16713R1

Predictors of self-reported symptoms and testing for COVID-19 in Canada using a nationally representative survey

PLOS ONE

Dear Dr. Jha,

Thank you for submitting your manuscript to PLOS ONE. After careful consideration, we feel that it has merit but does not fully meet PLOS ONE’s publication criteria as it currently stands. Therefore, we invite you to submit a revised version of the manuscript that addresses the points raised during the review process.

Reviewer 2 still has some minor clarification issues that should not be difficult to address.

We look forward to receiving your revised manuscript.

Kind regards,

Gabriel A. Picone

Academic Editor

PLOS ONE

Reviewers' comments:

Reviewer's Responses to Questions

**Comments to the Author**

1. If the authors have adequately addressed your comments raised in a previous round of review and you feel that this manuscript is now acceptable for publication, you may indicate that here to bypass the “Comments to the Author” section, enter your conflict of interest statement in the “Confidential to Editor” section, and submit your "Accept" recommendation.

Reviewer #1: All comments have been addressed

Reviewer #2: All comments have been addressed

2. Is the manuscript technically sound, and do the data support the conclusions?

Reviewer #1: Yes

Reviewer #2: Yes

3. Has the statistical analysis been performed appropriately and rigorously? 

Reviewer #1: Yes

Reviewer #2: Yes

4. Have the authors made all data underlying the findings in their manuscript fully available?

Reviewer #1: Yes

Reviewer #2: Yes

5. Is the manuscript presented in an intelligible fashion and written in standard English?

Reviewer #1: Yes

Reviewer #2: Yes

6. Review Comments to the Author

Reviewer #1: I do not have any further comments to the Authors. My earlier comments have been responded to and clarified.

Reviewer #2: The authors have painstakingly revised the manuscript in line with the review comments. However, the following revision will be required in order to make the manuscript publication worthy.

1. It is still unclear how the weighting of the sample size was done. This will require the authors to provide more details so as to bring clarity to it.

2. It is important for the authors to provide information on the source of the data collection instrument; was it adopted, adapted or developed for this study? Also, the method(s) of validating or assessing the reliability of the tool may be required. The statement on line 79 is unclear, which sampling frame were the authors referring to? The pretesting should not have been done among the same subjects who participated in the survey because it would have pre-empted them "The survey instrument was pre-tested in 60 individuals within the sampling frame."

3. It will also be good if the authors provide some sentence on the measures of used e.g. crude and adjusted odds ratio and the 95% confidence interval and rationale for using them in the data analysis section.

4. There is the need to provide footnotes for other ethnicity in tables 2 and 3.

7. PLOS authors have the option to publish the peer review history of their article (what does this mean?). If published, this will include your full peer review and any attached files.

Reviewer #1: **Yes: **Stanley Meribe, MD, PhD

Reviewer #2: **Yes: **Tolulope Olumide Afolaranmi

---

## [Author Response · Author response to Decision Letter 1]

22 Sep 2020

Comment 1- The authors have painstakingly revised the manuscript in line with the review comments. However, the following revision will be required in order to make the manuscript publication worthy.

It is still unclear how the weighting of the sample size was done. This will require the authors to provide more details so as to bring clarity to it.

Response- We have now added a line stating that “The sample frame was established to match the Canadian census data from Statistics Canada.” (lines 58-59) under the Study design section. We have now also mentioned that “Since the sample frame matched the Canadian census data from Statistics Canada, no additional survey weight was applied.” (lines 109-110).

Comment 2- It is important for the authors to provide information on the source of the data collection instrument; was it adopted, adapted or developed for this study? Also, the method(s) of validating or assessing the reliability of the tool may be required. The statement on line 79 is unclear, which sampling frame were the authors referring to? The pretesting should not have been done among the same subjects who participated in the survey because it would have pre-empted them "The survey instrument was pre-tested in 60 individuals within the sampling frame."

Response- We have now added a line on the source of the data collection instrument stating that the questionnaire used for this study was developed based on expert opinion (line 80). We have clarified that the 60 people were tested prior to the main survey (our apologies for the earlier error) (line 83). The data collection instrument (questionnaire) was not “validated” as at the time of survey, there were no reliable seroepidemiological studies that related infection with symptoms (line 235-237). However, subsequent studies comparing antibody with symptoms which we defined as suggestive of COVID in our study confirmed that our choice of symptoms was appropriate. We have also mentioned this in the Limitations section (line 237-239). 

Comment 3- It will also be good if the authors provide some sentence on the measures of used e.g. crude and adjusted odds ratio and the 95% confidence interval and rationale for using them in the Analysis section.

Response- We have now provided a sentence on the measure of outcome used (adjusted odds ratio and 95% confidence interval) and the rationale for using them in the Data analysis section (lines 106-108).

Comment 4- There is the need to provide footnotes for other ethnicity in tables 2 and 3. 

Response- We have now included footnotes for “Others” ethnicity in tables 1, 2, and 3 listing the list of included ethnicities.

---

## [Decision Letter · Decision Letter 2]

5 Oct 2020

Predictors of self-reported symptoms and testing for COVID-19 in Canada using a nationally representative survey

PONE-D-20-16713R2

Dear Dr. Jha,

We’re pleased to inform you that your manuscript has been judged scientifically suitable for publication and will be formally accepted for publication once it meets all outstanding technical requirements.

Kind regards,

Gabriel A. Picone

Academic Editor

PLOS ONE

Additional Editor Comments (optional):

Reviewers' comments:

Reviewer's Responses to Questions

**Comments to the Author**

1. If the authors have adequately addressed your comments raised in a previous round of review and you feel that this manuscript is now acceptable for publication, you may indicate that here to bypass the “Comments to the Author” section, enter your conflict of interest statement in the “Confidential to Editor” section, and submit your "Accept" recommendation.

Reviewer #2: All comments have been addressed

2. Is the manuscript technically sound, and do the data support the conclusions?

Reviewer #2: Yes

3. Has the statistical analysis been performed appropriately and rigorously? 

Reviewer #2: Yes

4. Have the authors made all data underlying the findings in their manuscript fully available?

Reviewer #2: Yes

5. Is the manuscript presented in an intelligible fashion and written in standard English?

Reviewer #2: Yes

6. Review Comments to the Author

Reviewer #2: The authors have revised the manuscript in line with all the review comments. I am positive that this manuscript is publication worthy and will be of interest to readers in the scientific community.

7. PLOS authors have the option to publish the peer review history of their article (what does this mean?). If published, this will include your full peer review and any attached files.

Reviewer #2: **Yes: **Tolulope Olumide Afolaranmi

---

## [Editor Report · Acceptance letter]

16 Oct 2020

PONE-D-20-16713R2 

Predictors of self-reported symptoms and testing for COVID-19 in Canada using a nationally representative survey 

Dear Dr. Jha:

I'm pleased to inform you that your manuscript has been deemed suitable for publication in PLOS ONE. Congratulations! Your manuscript is now with our production department. 

Kind regards, 

on behalf of

Dr. Gabriel A. Picone 

Academic Editor

PLOS ONE